# Trends of Alcohol Use, Dietary Behaviour, Interpersonal Violence, Mental Health, Oral and Hand Hygiene Behaviour among Adolescents in Lebanon: Cross-Sectional National School Surveys from 2005, 2011 and 2017

**DOI:** 10.3390/ijerph17197096

**Published:** 2020-09-28

**Authors:** Supa Pengpid, Karl Peltzer

**Affiliations:** 1ASEAN Institute for Health Development, Mahidol University, Salaya, Phutthamonthon, Nakhon Pathom 73170, Thailand; supaprom@yahoo.com; 2Department of Research Administration and Development, University of Limpopo, Polokwane 0727, South Africa; 3Department of Psychology, University of the Free State, Bloemfontein 9300, South Africa

**Keywords:** alcohol use, violence, injury, mental health, hygiene, protective factors

## Abstract

Health risk behaviours during adolescence can have long-term negative consequences. Little is known, however, about the recent health risk behaviour trends in adolescents in Lebanon. This investigation aimed to report the trends in the prevalence of various health risk behaviours, such as alcohol use, dietary behaviour, interpersonal violence, mental health, oral and hand hygiene, among adolescents in Lebanon. Cross-sectional nationally representative data were analysed from 13,109 adolescents (14 years median age) that participated in three waves (2005, 2011 and 2017) of the “Lebanon Global School-Based Student Health Survey (GSHS)”. Results indicate that significant improvements were found among both boys and girls in the decline in interpersonal violence (bulling victimization, being physically attack and involvement in physical fighting), poor washing of hands after using the toilet, and suicide planning, and among girls only loneliness, worry-induced sleep disturbance and suicidal ideation. Significant increases were found among both boys and girls in the prevalence of inadequate fruit consumption, and among boys only unintentional injury and not always washing hands before eating. In conclusion, several decreases but also increases in health risk behaviours were found over three assessment points during a period of 12 years calling for continued health enhancing activities in this adolescent population.

## 1. Introduction

Lebanon, an upper middle-income country in the Arab region, has a population of 5.5 million (2.4 million in Beirut) [1], including 1.3–1.8 million Syrian and Palestinian refugees [2]. The life expectancy at birth in Lebanon is 78.3 years, the urban population is 88.9%, most are Arab by ethnic group (95%), and 61.1% are Muslim and 33.7% Christian [1]. The various internal conflicts experienced by Lebanon as well as being the host for refugees of neighbouring countries have contributed to its social and economic instability [2].

In Lebanon, 91% of all deaths are attributed to non-communicable diseases (NCDs), including 47% cardiovascular diseases, 16% cancers, 6% injuries, 5% diabetes, 4% chronic respiratory diseases, and 19% other NCDs [3]. The burden of NCDs is on the increase in the Arab world, including Lebanon [4]. In some Arab countries, adolescents are disproportionally affected by cardiovascular and metabolic conditions, transport injuries, overweight/obesity, and mental health disorders, with tobacco use, unhealthy diets, physical inactivity, tobacco use, and violence are major risk factors [5,6]. For example, among school adolescents, compared to the global prevalence of combined physical behaviours (6.6%), the lowest prevalence was found in the Eastern Mediterranean region (4.9%) [7], and compared to the global prevalence of past-month bullying victimisation (34.4%), a higher prevalence was found in the Eastern Mediterranean region (38.2%) [8].

As stated by the World Health Organization (WHO), “alcohol use, dietary behaviours, drug use, hygiene, mental health, physical activity, protective factors, sexual behaviours, tobacco use, violence and unintentional injury” are the leading causes of morbidity/mortality among children and adults globally [9]. The global school-based student health survey (GSHS) is a World Health Organization collaborative surveillance project to measure and assess the behavioural NCD risk factors and protective factors focusing on students mainly aged 11–18 years old, and is collected every 4–6 years [9].

Previous cross-sectional studies among adolescents in Lebanon found a high prevalence of various health risk behaviours. In terms of interpersonal violence, a high prevalence of involvement in physical fight (almost 42%) were found among school adolescents in Beirut, Lebanon [10]. In a national sample of adolescents in Lebanon in 2009, 76.4% had experienced physical abuse at school [11], and in a study in 8- to 17-year-old Lebanese children, 54% reported past year physical abuse [12], and 30% were involved in bullying [13]. In a household survey in 2015 in Lebanon, 57% of Lebanese and 65% of Syrian children had been exposed to some type of violent discipline [2]. Regarding mental health, in a study among adolescents (N = 510) in Beirut, the prevalence of suicidal ideation or attempt was 4.3% [14], 76.5% had poor sleep quality [15], and 26.1% had 30 day psychiatric disorders, including 13.1% who had anxiety disorders [16]. In a national survey among adolescents in Lebanon, a high prevalence of household food insecurity (55.2%) was observed [17], and in a survey among adolescent students (N = 830) in Beirut, Lebanon, in 2014, a high prevalence of sub-optimal tooth brushing (54.0%, <2 times/day) was found [18]. Considering the high burden of NCDs, its risk factors and other health risk behaviours in Lebanon, a major research gap is to investigate behavioural risk factors of NCDs and other health risk behaviours among youth over time in Lebanon.

Assessing behavioural risk factors, such as injury and violence, poor mental health, substance use, unhealthy diet and sedentary lifestyle, over time may aid in improving strategies to prevent or ameliorate reducing health risk behaviours in the adolescent population [19,20,21]. Health education has been integrated into the school curriculum in Lebanon since 1987 [22], and several upgrades of the school health programme have taken place in Lebanon since then [23]. Several evaluations of the school health programme in Lebanon have been conducted with mixed results. In a survey of 50 principals from 50 different schools ranging from middle to secondary schools selected from various areas in 2013 in Lebanon, 80% of the school principals reported that their schools do not offer any health educations to their students [24]. In another study including 50 schools in Lebanon, 70% of schools provided health-related courses [22]. In a cross-sectional evaluation comparing ten health-promoting schools (HPS) in Lebanon in 2010 with non-HPS, no significant differences between HPS and non-HPS in the assessed risk behaviours (drug use, smoking and alcohol use) were found [23].

Studies investigating trends in health risk behaviours among adolescents showed different results [20,21,25], ranging from increases in the prevalence of fruit and vegetable intake and in being bullied to decreases in physical inactivity, bullying victimization, fighting and injury. In a trend study in Oman, poor hand hygiene behaviour increased [21], while it decreased in the Philippines [20]. A major gap in the literature is the lack of studies that have assessed trends in health risk behaviour among adolescents in countries in the Arab region, such as in Lebanon. The current investigation assessed trends of the prevalence of 19 different health risk behaviours and protective factors in the 2005, 2010 and 2017 Lebanon GSHS. It is hypothesized that the proportion of the various unhealthy behaviours assessed would be different across the three cross-sectional survey assessments over a period of 12 years. Study findings on various health risk behaviours over time could be helpful in strategies promoting health programme activities in schools [26].

## 2. Methods

### 2.1. Sample and Procedure

Cross-sectional nationally representative data from the 2005, 2011 and 2017 Lebanon GSHS were analysed [9]. More detailed information on the methods of the GSHS and the data can be publicly accessed [9]. Briefly, a two-stage cluster sampling design was used to generate a national representative country sample [9]. First, schools were selected with probability proportional to sample size. Second, classes of students in grades 7–9 in 2005 and 2011 and grades 7–12 in 2017 within schools were randomly selected [9]. All students in the selected classes were eligible to participate irrespective of age [9]. Data were collected with a self-administered questionnaire in Arabic language [9]. For the 2005 Lebanon GSHS, the school response rate was 92%, the student response rate was 96%, and the overall response rate was 88%; for the 2011 Lebanon GSHS 2011, the corresponding figures were 88%, 99% and 87%, and for the 2017 Lebanon GSHS the corresponding figures were 88%, 94% and 82%, respectively [9]. “A national ethics committee approved the study and written informed consent was obtained from the participating schools, parents and students.” [9].

### 2.2. Measures

The questionnaire used is shown in Table 1 [9]. The GSHS core questionnaire assesses 10 modules: “Alcohol use, Dietary behaviors, Drug use, Hygiene, Mental health, Physical activity, Protective factors, Sexual behaviors that contribute to HIV infection, other sexually-transmitted infections, and unintended pregnancy, Tobacco use, Violence and unintentional injury.” [9] All GSHS core modules that were assessed in the 2005, 2011 and 2017 Lebanon GSHS were included in this analysis. This included dietary behaviour (fruit and vegetable intake, and the experience of hunger), alcohol use (current use, ever drunk, and trouble from alcohol use), injury and violence (past 12 month injury, being bullied in the past month, involvement in a physical fight in the past year and having been physically attacked in the past year), oral and hand hygiene (brushing teeth, hand washing before eating, after toilet and using soap when hand washing), poor mental health (having no close friends, loneliness, worry-induced sleep disturbance, suicidal ideation and suicide attempt in the past year), and protective factors (school attendance, peer support, parental supervision, connectedness and bonding) (see Table 1). The consumption of less than “two or more servings of fruits in a day” and less than “three or more servings of vegetables a day” were considered inadequate [27].

### 2.3. Data Analysis

Statistical analyses were done with “STATA software version 15.0 (Stata Corporation, College Station, TX, USA)”. Data from the 2005, 2011 and 2017 Lebanon GSHS were merged and were weighted for non-response and probability selection [9]. Descriptive data were presented as proportions for each study year. Moreover, the absolute differences of each health risk behaviour outcome between the first and last study year were shown in percentages. Pearson chi-square tests were applied for testing differences in proportion. The significance of a linear trend was tested by treating study year as categorical in logistic regression analyses, controlled for age group and experience of hunger (proxy for socioeconomic status), for boys and girls, separately. Since alcohol use, dietary behaviour, interpersonal violence, mental health, oral and hand hygiene behaviour may differ by sex, we thought it being more appropriate to conduct sex stratified analysis, in line with some previous studies [20,21,25]. Taylor linearization methods were utilized in all statistical analyses to account for the sample weight and complex design of the study. Under 3.5% of the data were missing for all the variables (except for injury). *p* < 0.05 was considered significant. Missing values were excluded from the analysis.

## 3. Results

### 3.1. Study Sample Description

The three Lebanon GSHS samples consisted of 13,109 school-going adolescents, 53.0% females and 47.0% males (14 years median age, 2 years interquartile range). The proportion older adolescents increased across the three different surveys X^2^ (6, N = 13,059) = 1083.26, *p* ≤ 0.001) (see Table 2).

### 3.2. Health Risk Behaviour Outcomes

#### 3.2.1. Poor Diet

Among students, 42.2% of males and 47.2% of females ate less than two fruit servings daily in 2005, and the proportion of inadequate fruit intake significantly increased to 50.3% among boys (*p* for trend 0.006) and 54.4% among girls (*p* for trend 0.007) in 2017. Inadequate vegetable consumption (81.4% among boys and 85.7% among girls in 2005) did not significantly change over time among both boys and girls. The prevalence of students who reported frequently experiencing hunger (3.4% among boys and 2.6% among girls in 2005) did not significantly change from 2005 to 2017 in both males and females.

#### 3.2.2. Alcohol Use and Misuse

The prevalence of current alcohol use (27.8% among boys and 12.2% among girls in 2005), ever drunk (21.2% among boys and 7.1% among girls in 2005) and trouble from drinking alcohol (23.7% among boys and 11.1% among girls in 2005) did not significantly change over time among both boys and girls.

#### 3.2.3. Injury and Violence-Related Behaviour

The prevalence of annual injury increased significantly from 37% to 44% in boys (*p* for trend 0.004) but not in girls (28.4% in 2005 and 31.5% in 2017). Being bullied (38.7% among boys and 29.4% among girls in 2005), a victim of physical assault (50.0% among boys and 33.4% among girls in 2005) and involved in physical fighting (64.6% among boys and 29.0% among girls in 2005) significantly decreased in both sexes (*p* for trend <0.001).

#### 3.2.4. Oral and Hand Hygiene

The prevalence of inadequate tooth brushing was 38.4% among male and 30.4% among female students in 2005, which remained high over time. Not always washing hands prior to eating significantly increased among boys (*p* for trend 0.012), while not always washing hands after using the toilet decreased both among boys (*p* for trend 0.002) and girls (*p* for trend <0.001), and “not always washing hands with soap” did not change over time.

#### 3.2.5. Poor Mental Health

Among all five poor mental health indications (having no close friends: 3.7% among boys and 3.3% among girls in 2005, worry-induced sleep disturbance: 9.3% among boys and 17.7% among girls in 2005, loneliness: 7.7% among boys and 16.1% among girls in 2005, suicidal ideation: 14.5% among boys and 17.5% among girls in 2005, and suicide planning: 11.0% among boys and 11.2% among girls in 2005), suicide planning decreased among boys (*p* for trend <0.001), and loneliness (*p* for trend 0.004), worry-induced sleep disturbance (*p* for trend 0.022), suicidal ideation (*p* for trend <0.001) and suicide planning (*p* for trend <0.001) significantly decreased among girls.

#### 3.2.6. Protective Factors

School truancy (20.9% among boys and 10.3% among girls in 2005), peer support (64.5% among boys and 73.3% among girls in 2005) and the three parental support indicators (supervision, connectedness and bonding) did not change over time among both boys and girls (see Table 3 and Table 4).

## 4. Discussion

The study found, across three GSHS in 2005, 2011 and 2017 in Lebanon, significant decreases among both boys and girls in interpersonal violence (bulling victimization, being physically attacked and involvement in physical fighting), inadequate hand hygiene (after toilet use), and suicide planning, and among girls only suicidal ideation, loneliness and worry-induced sleep disturbance. Significant increases were found among both boys and girls in the prevalence of inadequate fruit consumption, and among boys only unintentional injury and inadequate hand hygiene (before eating). Several studies investigating the implementation of health education in schools in Lebanon showed poor or mixed results. For example, in a survey of 50 different schools, 80% of the school principals reported that their schools do not offer any health educations to their students [24], and in another study involving 50 schools from different areas of Lebanon, that the majority of schools had either implemented or were in the process of implementing a health promoting school programme, and the most popular health related courses offered included dental health (74%), smoking cessation (72%), physical activity (68%), and mental health (20%) [22]. However, a cross-sectional evaluation comparing HPS with non-HPS found no significant differences between HPS and non-HPS in relation to drug use, smoking and alcohol use [23].

Violence-related behaviour (being bullied, physically attacked and participation in a physical fight) decreased in this study, which concurs with four other studies [19,28,29,30], while a few studies found an increase in one or more types of interpersonal violence, e.g., in Oman [21], the Philippines [20] and Venezuela [31]. In several older studies among adolescents in Lebanon, high rates of interpersonal violence have been reported [10,11,12], which aligns with our high rates of interpersonal violence in the 2005 GSHS. It is possible that the high rates of interpersonal violence in 2005 were still related to the post-conflict situation in Lebanon, which subsequently subsided so that interpersonal violence decreased from 2005 to 2017. On the other hand, injury prevalence increased in this study among boys, while in the trend study in the Philippines, a similar increase was observed [20], in Oman, no significant trend differences were found [21] and in Morocco, a decline in the prevalence of injury among adolescents was found [32]. Considering the high proportion and significant increase in annual injury prevalence in this study intensified safety promotion and injury prevention programming is indicated in Lebanon.

The prevalence of inadequate fruit and vegetable intake was high in the 2005 GSHS and increased for fruit intake over the study period, which was also shown in a trend study in Oman [21] and other countries in the Arab region [33]. The experience of hunger (or food insecurity) was low and did not significantly differ among boys and girls over time. In a national survey among adolescents conducted in 2015 in Lebanon, a high prevalence of household food insecurity (55.2%) was observed [17], which may explain the high prevalence of food insecurity in our 2011 survey.

The prevalence of sub-optimal oral hygiene (tooth brushing <twice/day) was high across the three school surveys (almost 40%), much higher than in a study among school adolescents in four Southeast Asian countries (22.4%) [34]. In a survey among adolescent students in Beirut, a high prevalence of sub-optimal tooth brushing was also found [18], calling for oral hygiene health promotion programmes targeting schoolchildren and their parents [18]. Although poor hand washing after toilet use decreased among both sexes, poor hand washing before eating increased among boys in this study. In the Oman trend study, sub-optimal hand hygiene pattern increased [21], while it decreased in the Philippines [20]. It is possible that poor hand washing after toilet use decreased among adolescents in Lebanon from 2011, after the “Call to Action for WASH in Schools campaign was formally launched in 2010” in Lebanon [35].

Regarding mental health indicators (having no close friends, loneliness, worry-induced sleep disturbance, suicidal ideation and suicide plan), suicide planning decreased among both boys and girls, while loneliness, suicidal ideation and worry-induced sleep disturbance decreased among girls only. In comparison, in the Philippines trend study, the prevalence of suicidal ideation and suicide planning decreased among boys and suicidal ideation increased among girls over time [20]. As found in previous investigations [14,15,16], poor mental health, such as suicidal behaviour and anxiety-related disturbances, has been identified as a significant problem among adolescents in Lebanon.

In terms of protective factors, school attendance, peer support and the three parental support indicators (supervision, connectedness and bonding) did not change over time among both boys and girls. In the Philippines trend study, protective factors did not change over time [20], in the Oman trend study only one of the protective factors (peer support) improved over time [21], while in New Zealand trend study, positive family and school connections improved over time [19].

The current study results may inform public health intervention programmes targeting specific health risk behaviours among adolescents in Lebanon. For example, specific school food environment policies, such as direct provision of healthful foods/beverages, can improve targeted dietary behaviours, such as fruit and vegetable intake [36]. Universal school-based interventions that target multiple risk behaviours may be effective in preventing engagement in substance use, including alcohol use, among young people [37]. There is “good evidence that various whole-school health interventions are effective in preventing bullying.” [38]. School dental health education can improve oral hygiene practice behaviours, such as frequency and duration of brushing, of school children [39,40]. Increased implementation of multi-level (training, funding and policy) “hand-washing interventions can reduce the incidence of diarrhoea, respiratory infections, and school absenteeism.” [41]. Universal resilience-focused interventions (particularly cognitive-behavioural therapy-based approaches) may be used for reductions in poor mental health (depressive and anxiety symptoms) for children and adolescents [42].

### Limitations of the Study

The “Secondary education enrolment ratio” was 80% in Lebanon in 2005, 76% in 2011 and 63% in 2017 [43], yet this school survey did not represent all adolescents in Lebanon. Some study variables (such as sedentary behaviour, tobacco use, sexual behaviour, and physical activity) were not included in this analysis, since they had not been assessed in all the three Lebanon GSHS. Although self-reported weight and height were collected, the missing values were in the wave one survey more than 30%, and therefore body mass index was not included in this report. Furthermore, the study is limited because of its cross-sectional design and self-reported data collection.

## 5. Conclusions

In this investigation of nationally representative school adolescents over a period of 12 years in Lebanon, significant improvements were found among both boys and girls in the decline of interpersonal violence (bulling victimization, being physically attacked and involvement in physical fighting), inadequate hand hygiene (after toilet use), and suicide planning, and among girls only loneliness, worry-induced sleep disturbance and suicidal ideation. Significant increases were found among both boys and girls in the prevalence of inadequate fruit consumption, and among boys only unintentional injury and inadequate hand hygiene (before eating). Several decreases but also increases in health risk behaviours were identified over three cross-sectional surveys from 2005 to 2017 calling for continued interventions in promoting health behaviour in this adolescent population.

## Figures and Tables

**Table 1 ijerph-17-07096-t001:** Variable description.

Variables	Question	Response Options (Coding Scheme)
Age	“How old are you?”	“11 years old or younger to 16 or 18 years old or older”
Sex	“What is your sex?”	“Male, Female”
Grade	“In what grade are you?”	
	**Dietary behaviour**	
Fruits	“During the past 30 days, how many times per day did you usually eat fruit such as apples, bananas, and oranges?”	“1 = I did not eat fruit during the past 30 days to 7 = 5 or more times per day (coded 1–3 = 1 and 4–8 = 0)”
Vegetables	“During the past 30 days, how many times per day did you usually eat vegetables, such as salads, spinach, eggplant, tomatoes, and cucumbers?”	“I did not eat vegetables during the past 30 days to 7 = 5 or more times per day (coded 1–4 = 1 and 5–8 = 0”
Hunger	“During the past 30 days, how often did you go hungry because there was not enough food in your home?”	“1 = never to 5 = always (coded 1–3 = 0 and 4–5 = 1)”
	**Alcohol use**	
Current alcohol use	“During the past 30 days, on how many days did you have at least one drink containing alcohol?”	“1 = 0 days to 7 = All 30 days (coded 1 = 0 and 2–7 = 1)”
Ever drunk	“During your life, how many times did you drink so much alcohol that you were really drunk?”	“1 = 0 times and 4 = 10 or more times (coded 1 = 0 and 2–4 = 1)”
Trouble from alcohol use	“During your life, how many times have you got into trouble with your family or friends, missed school, or got into fights, as a result of drinking alcohol?”	“1 = 0 times and 4 = 10 or more times (coded 1 = 0 and 2–4 = 1)”
	**Injury and violence**	
Injury	“During the past 12 months, how many times were you seriously injured?”	1 = 0 times to 8 = 12 or more times (coded 1 = 0 and 2–8 = 1)
Bullied	“During the past 30 days, on how many days were you bullied?”	“1 = 0 days to 7 = All 30 days (coded 1 = 0 and 2–7 = 1)”
In a physical fight	“During the past 12 months, how many times were you in a physical fight?”	“1 = 0 times to 8 = 12 or more times (coded 1 = 0 and 2–8 = 1)”
Physically attacked	“During the past 12 months, how many times were you physically attacked?”	“1 = 0 times to 8 = 12 or more times (coded 1 = 0 and 2–8 = 1)”
	**Oral and hand hygiene**	
Brushing teeth (≤1 time/day)	“During the past 30 days, how many times per day did you usually clean or brush your teeth?”	“1 = never to 6 = 4 or more times a day (coded 1–3 = 1 and 4–6 = 0)”
Hand washing before eating	During the past 30 days, how often did you wash your hands before eating?	“1 = never to 5 = always (coded 1–4 = 1 and 5 = 0)”
Hand washing after toilet	During the past 30 days, how often did you wash your hands after using the toilet or latrine?	“1 = never to 5 = always (coded 1–4 = 1 and 5 = 0)”
Using soap when hand washing	During the past 30 days, how often did you use soap when washing your hands?	“1 = never to 5 = always (coded 1–4 = 1 and 5 = 0)”
	**Poor mental health**	
No close friends	“How many close friends do you have?”	“1 = 0 to 4 = 3 or more (coded 1+ = 0, 0 = 1)”
Loneliness	“During the past 12 months, how often have you felt lonely?”	“1 = never to 5 = always (coded 1–3 = 0 and 4–5 = 1)”
Worry-induced sleep disturbance	“During the past 12 months, how often have you been so worried about something that you could not sleep at night?”	“1 = never to 5 = always (coded 1–3 = 0 and 4–5 = 1)”
Suicide ideation	“During the past 12 months, did you ever seriously consider attempting suicide?”	“Yes, No”
Suicide attempt	“During the past 12 months, how many times did you actually attempt suicide?”	“1 = 0 times to 5 = 6 or more times (coded 1 = 0 and 2–5 = 1)”
	**Protective factors**	
School truancy	During the past 30 days, on how many days did you miss classes or school without permission?	“1 = 0 days to 5 = 10 or more days (coded 1 = 0 and 2–5 = 1)”
Peer support	“During the past 30 days, how often were most of the students in your school kind and helpful?”	“1 = never to 5 = always (coded 1–3 = 0 and 4–5 = 1)”
Parental supervision	“During the past 30 days, how often did your parents or guardians check to see if your homework was done?”	“1 = never to 5 = always (coded 1–3 = 0 and 4–5 = 1)”
Parental connectedness	“During the past 30 days, how often did your parents or guardians understand your problems and worries?”	“1 = never to 5 = always (coded 1–3 = 0 and 4–5 = 1)”
Parental bonding	“During the past 30 days, how often did your parents or guardians really know what you were doing with your free time?	“1 = never to 5 = always (coded 1–3 = 0 and 4–5 = 1)”

**Table 2 ijerph-17-07096-t002:** Characteristics of participating students in the 2005, 2011 and 2017 surveys in Lebanon (N = 13,109).

Variable	2005 (N = 5115)	2011 (N = 2286)	2017 (N = 5708)
	N (%)	N (%)	N (%)
Gender			
Male	2333 (47.7)	1064 (46.7)	2330 (46.8)
Female	2776 (52.3)	1220 (53.3)	3370 (53.2)
Missing	16	27	8
Age in years			
13 years or younger	2211 (43.1)	910 (38.5)	1543 (31.3)
14	1341 (26.1)	619 (28.6)	953 (19.1)
15	1020 (20.3)	477 (22.6)	925 (15.9)
16 years or older	522 (10.5)	267 (10.3)	2271 (33.7)
Missing	21	9	6
Grade			
7	1988 (38.8)	922 (38.1)	1247 (23.5)
8	1578 (32.1)	910 (32.7)	1125 (20.0)
9	1540 (29.1)	452 (29.2)	783 (17.1)
10–12	0	0	2526 (39.4)
Missing	13	2	2

**Table 3 ijerph-17-07096-t003:** Health risk behaviours in 2005, 2011 and 2017 among male in-school adolescents in Lebanon.

Variable	2005	2011	2017	Difference ^1^	*p*-for Trend ^2^
	N (%)	N (%)	N (%)	%	
**Dietary behaviour**					
Fruits <2 servings/day	989 (42.2)	463 (45.4)	1152 (50.3)	+8.1	0.006
Vegetable <3 servings/day	1890 (81.4)	820 (76.6)	1888 (82.0)	+0.6	0.970
Went hungry (mostly/always) in the past 30 days	78 (3.4)	57 (5.1)	71 (3.4)	+0.0	0.959
**Alcohol use**					
Current alcohol use (past month)	637 (27.8)	320 (35.7)	497 (22.9)	−4.9	0.106
Ever drunk	490 (21.2)	239 (25.9)	381 (17.4)	−3.7	0.055
Trouble from alcohol use (lifetime)	540 (23.7)	70 (8.2)	366 (19.3)	−4.4	0.198
**Injury and violence**					
Any serious injury (past 12 months)	622 (37.0)	413 (44.0)	865 (43.5)	+6.5	0.004
Bullied (past months)	790 (38.7)	331 (33.3)	438 (21.0)	−17.7	<0.001
In physical fight (past 12 months)	1493 (64.6)	742 (70.2)	1231 (55.1)	−9.5	<0.001
Physically attacked (past 12 months	1137 (50.0)	512 (46.0)	577 (25.0)	−25.0	<0.001
**Oral and hand hygiene**					
Brushing teeth (≤once/day/past 30 days)	897 (38.4)	415 (41.9)	959 (39.4)	+1.0	0.579
Wash hands before eating (not always) (past 30 days)	624 (26.4)	314 (32.4)	774 (32.9)	+6.5	0.012
Wash hands after toilet/latrine use (not always) (past 30 days)	364 (15.7)	120 (11.6)	262 (10.7)	−5.0	0.002
Wash hands with soap (not always) (past 30 days)	585 (25.5)	271 (25.1)	565 (22.4)	−3.1	0.066
**Poor mental health**					
Having no close friends	85 (3.7)	42 (4.0)	90 (3.5)	−0.2	0.581
Loneliness (past 12 months)	436 (7.7)	197 (8.6)	536 (8.2)	+0.5	0.992
Worry-induced sleep disturbance (past 12 months)	214 (9.3)	88 (8.9)	233 (9.6)	+0.6	0.629
Suicidal ideation (past 12 months)	333 (14.5)	125 (12.1)	277 (12.7)	−1.8	0.193
Suicide plan (past 12 months	248 (11.0)	104 (10.2)	176 (7.8)	−3.2	<0.001
**Protective factors**					
Truancy (past 30 days)	462 (20.9)	247 (22.5)	464 (20.2)	−0.7	0.301
Peer support (mostly/always) (past 30 days)	1489 (64.5)	644 (65.5)	1305 (64.3)	−0.2	0.536
Parents/guardians supervision (mostly/always) (past 30 days)	1168 (51.2)	518 (47.4)	949 (47.0)	−4.2	0.539
Parents/guardians connectedness (mostly/always) (past 30 days)	1058 (46.0)	471 (45.4)	953 (46.6)	+0.6	0.423
Parents or guardians bonding (mostly/always) (past 30 days)	1014 (44.3)	500 (49.1)	826 (40.9)	−3.4	0.201

^1^ Difference between 2005 and 2017; ^2^
*p* for trend is controlled for age group and experience of hunger (as proxy for socioeconomic status).

**Table 4 ijerph-17-07096-t004:** Health risk behaviours in 2005, 2011 and 2017 among female in-school adolescents in Lebanon.

Variable	2005	2011	2017	Difference ^1^	*p*-for Trend ^2^
	N (%)	N (%)	N (%)	%	
**Dietary behaviour**					
Fruits <2 servings/day	1301 (47.2)	629 (52.8)	1835 (54.4)	+7.2	0.007
Vegetable <3 servings/day	2353 (85.7)	986 (81.2)	2873 (84.7)	−1.0	0.535
Went hungry (mostly/always) in the past 30 days	71 (2.6)	38 (2.7)	91 (3.2)	+0.6	0.629
**Alcohol use**					
Current alcohol use (past month)	329 (12.2)	222 (20.3)	319 (12.8)	+0.6	0.960
Ever drunk	192 (7.1)	167 (15.8)	206 (8.1)	+1.0	0.887
Trouble from alcohol use (lifetime)	300 (11.1)	27 (2.1)	234 (9.6)	−1.5	0.817
**Injury and violence**					
Any serious injury (past 12 months)	594 (28.4)	382 (35.3)	915 (31.5)	+3.1	0.649
Bullied (past months)	736 (29.4)	191 (16.6)	371 (12.6)	−16.8	<0.001
In physical fight (past 12 months)	798 (29.0)	357 (30.2)	728 (23.5)	−5.5	<0.001
Physically attacked (past 12 months	911 (33.4)	426 (35.6)	550 (16.6)	−16.8	<0.001
**Oral and hand hygiene**					
Brushing teeth (≤once/day/past 30 days)	839 (30.4)	342 (29.9)	1129 (30.9)	+0.5	0.890
Wash hands before eating (not always) (past 30 days)	751 (27.3)	377 (31.2)	1073 (31.1)	+3.8	0.525
Wash hands after toilet/latrine use (not always) (past 30 days)	321 (11.6)	113 (10.1)	224 (7.5)	−4.1	<0.001
Wash hands with soap (not always) (past 30 days)	509 (18.6)	187 (15.5)	609 (16.6)	−2.0	0.065
**Poor mental health**					
Having no close friends	90 (3.3)	45 (3.3)	178 (4.6)	+1.3	0.204
Loneliness (past 12 months)	436 (16.1)	197 (16.4)	536 (15.2)	−0.9	0.004
Worry-induced sleep disturbance (past 12 months)	483 (17.7)	194 (14.5)	605 (17.2)	−0.5	0.022
Suicidal ideation (past 12 months)	471 (17.5)	209 (17.8)	487 (14.1)	−3.4	<0.001
Suicide plan (past 12 months	299 (11.2)	158 (12.9)	322 (9.1)	−2.1	<0.001
**Protective factors**					
Truancy (past 30 days)	281 (10.3)	176 (15.1)	432 (13.3)	+3.0	0.390
Peer support (mostly/always) (past 30 days)	2002 (73.3)	818 (72.2)	2248 (72.8)	−0.5	0.615
Parents/guardians supervision (mostly/always) (past 30 days)	1303 (47.4)	505 (42.2)	1393 (45.5)	−1.9	0.206
Parents/guardians connectedness (mostly/always) (past 30 days)	1330 (48.7)	555 (49.4)	1504 (49.4)	+0.7	0.113
Parents or guardians bonding (mostly/always) (past 30 days)	1427 (52.3)	616 (56.3)	1652 (54.5)	+2.2	0.103

^1^ Difference between 2005 and 2017; ^2^
*p* for trend is controlled for age group and experience of hunger (as proxy for socioeconomic status).

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
