# Peer review of "Trends of Alcohol Use, Dietary Behaviour, Interpersonal Violence, Mental Health, Oral and Hand Hygiene Behaviour among Adolescents in Lebanon: Cross-Sectional National School Surveys from 2005, 2011 and 2017"

_ijerph, 2020, doi:10.3390/ijerph17197096_

Round 1

Reviewer 1 Report

This manuscript presents an analysis of cross-sectional cohort data from Lebanon school children. Overall, the sample is quite large and there are extensive behaviors available, which are strengths. However, I have a number of questions/concerns about the framing, background, and analysis for the authors to consider.

  1. There are a number of important details about the context of the study, which are not introduced until the discussion section. As well, the background does not currently set up why looking at all these behaviors and looking at them over time in Lebanon is an important area of research. (Yes, there appears to be a gap, but the argument is lacking why this gap is important to address). Regarding the context – what is important to know about these health behaviors and risk behaviors in Lebanon school children specifically? What about the conflict/post-conflict situation in Lebanon specifically is important to understand. How does the introduction of the health promoting school fit into this picture?
  2. The authors appear to use extensive quotes from other documents. While it does not appear to be plagiarism as the quotes all have citations, this makes for an awkward reading experience. As well, there is a need for careful editing for grammar. Many sections may need a full rewrite as a result of these issues.
  3. The measures section should be expanded to discuss what the actual measures were and/or include a table with this information; some variables are mentioned later but they should be described here (i.e., experience of hunger which was used as a proxy for SES)
  4. From an analysis perspective, the introduction of the health promoting school network is important to account for as a potential confound in the analysis. I would have expected that the authors would analyze the data before and then after to see whether or not the schools made a difference (and report these findings). They seem to have done this in part in the discussion, but I wondered why this was not included in the manuscript as a formal part of their analysis plan. It seems the trends would be somewhat meaningless without accounting for this.
  5. Lines 87-88; it is unclear what was “inadequate” about the consumption of fruits and vegetables. This statement should be clarified.
  6. The results are quite difficult to get through as they are very long sentences that discuss the findings for both genders and across years. One potential option to clarify might be to split the findings into a discussion of the changes within gender across years and then to have a summary comparison of the genders at the end.
  7. Finally, I am not convinced that the focus on simply describing the behaviors over time and between genders is able to truly inform intervention efforts. One way to further contribute would be to examine whether certain healthy/unhealthy behaviors (and protective factors) occur together. I realize this is a new direction for the manuscript and would likely require adding some relevant theoretical framing to the background. I do think this direction would improve the implications of the study.

Author Response

Reviewer I:

This manuscript presents an analysis of cross-sectional cohort data from Lebanon school children. Overall, the sample is quite large and there are extensive behaviors available, which are strengths. However, I have a number of questions/concerns about the framing, background, and analysis for the authors to consider.

1. There are a number of important details about the context of the study, which are not introduced until the discussion section. As well, the background does not currently set up why looking at all these behaviors and looking at them over time in Lebanon is an important area of research. (Yes, there appears to be a gap, but the argument is lacking why this gap is important to address). Regarding the context – what is important to know about these health behaviors and risk behaviors in Lebanon school children specifically? What about the conflict/post-conflict situation in Lebanon specifically is important to understand. How does the introduction of the health promoting school fit into this picture?
Response: More is added in several places in the introduction to address above issues
2. The authors appear to use extensive quotes from other documents. While it does not appear to be plagiarism as the quotes all have citations, this makes for an awkward reading experience. As well, there is a need for careful editing for grammar. Many sections may need a full rewrite as a result of these issues.
Response: Corrected
3. The measures section should be expanded to discuss what the actual measures were and/or include a table with this information; some variables are mentioned later but they should be described here (i.e., experience of hunger which was used as a proxy for SES)
Response: The supplementary file 1 shows the questionnaire used, now this is changed to Table 1, as suggested
In addition, individual items of the measure are described, as below
All GSHS core modules that were assessed in the 2005, 2011 and 2017 Lebanon GSHS were included in this analysis. This included dietary behaviour (fruit and vegetable intake, and the experience of hunger) alcohol use (current use, ever drunk, trouble from alcohol use), injury and violence (past 12-month injury, being bullied in the past month, involvement in a physical fight in the past year and having been physically attacked in the past year), oral and hand hygiene (brushing teeth, hand washing before eating, after toilet and using soap when hand washing), poor mental health (no close friends, loneliness, worry-induced sleep disturbance, suicidal ideation and suicide attempt in the past year), and protective factors (school attendance, peer support, parental supervision, connectedness and bonding) (see Table 1).
4. From an analysis perspective, the introduction of the health promoting school network is important to account for as a potential confound in the analysis. I would have expected that the authors would analyze the data before and then after to see whether or not the schools made a difference (and report these findings). They seem to have done this in part in the discussion, but I wondered why this was not included in the manuscript as a formal part of their analysis plan. It seems the trends would be somewhat meaningless without accounting for this.
Response: We report about an evaluation of a pilot of 10 health promoting schools. We have no information which schools surveyed had implemented health promotion, also in the survey questionnaire no data were available on the exposure to health education. Therefore, we could not include a before and after study design, or control for exposure to health promotion.
5. Lines 87-88; it is unclear what was “inadequate” about the consumption of fruits and vegetables. This statement should be clarified.
Response: The reference is provided on the definition of insufficient or inadequate fruit or vegetables intake.
6. The results are quite difficult to get through as they are very long sentences that discuss the findings for both genders and across years. One potential option to clarify might be to split the findings into a discussion of the changes within gender across years and then to have a summary comparison of the genders at the end.
Response: Very long sentences have been broke up.
7. Finally, I am not convinced that the focus on simply describing the behaviors over time and between genders is able to truly inform intervention efforts. One way to further contribute would be to examine whether certain healthy/unhealthy behaviors (and protective factors) occur together. I realize this is a new direction for the manuscript and would likely require adding some relevant theoretical framing to the background. I do think this direction would improve the implications of the study.
Response: We have tried to group some of the indicators, such as fruit and vegetable intake, interpersonal violence or poor mental health, but in this trend analysis, we feel it is more relevant to report on individual indicators. This will help in better targeting interventions.

Reviewer 2 Report

About "introduction"

A more extensive introduction gathering general data on health (WHO) and teenagers (UNICEF), as well as related articles, is missing

WHO
https://www.who.int/countries/lbn/en/

UNICEF
https://www.unicef.org/lebanon/

World Health Organization, & Centers for Disease Control and Prevention (CDC. (2013). Global school-based student health survey (GSHS).

Han, L., You, D., Gao, X., Duan, S., Hu, G., Wang, H., ... & Zeng, F. (2019). Unintentional injuries and violence among adolescents aged 12–15 years in 68 low-income and middle-income countries: a secondary analysis of data from the global school-based student health survey. The Lancet Child & Adolescent Health3(9), 616-626.

Xu, G., Sun, N., Li, L., Qi, W., Li, C., Zhou, M., ... & Han, L. (2020). Physical behaviors of 12-15 year-old adolescents in 54 low-and middle-income countries: results from the Global School-based Student Health Survey. Journal of global health10(1).

Presentation of results
- The value p must be written in lowercase and italics
- In the tables, asterisks of significance could be added
*, p < 0.05; **, p < 0.01; ***, p < 0.001

- Chi-Square test of Independence:  degrees of freedom and sample size in parentheses, then the chi-square value, followed by the significance level. For example:
“Animal response to the stimuli did not differ by species, Χ2(1, N = 75) = 0.89, p = .25.”

In the discussion you could compare the results with other studies in surrounding countries (see references above)

In the conclusion you could add proposals for social and health policies to reverse certain trends

There are some mistakes:

line 156: "contolled" should be "controlled"

line 238: "respresentative" should be "representative" 

Author Response

Reviewer II
About "introduction"
A more extensive introduction gathering general data on health (WHO) and teenagers (UNICEF), as well as related articles, is missing
WHO
https://www.who.int/countries/lbn/en/
Response: added
UNICEF
https://www.unicef.org/lebanon/
Response: added
World Health Organization, & Centers for Disease Control and Prevention (CDC. (2013). Global school-based student health survey (GSHS).
Han, L., You, D., Gao, X., Duan, S., Hu, G., Wang, H., ... & Zeng, F. (2019). Unintentional injuries and violence among adolescents aged 12–15 years in 68 low-income and middle-income countries: a secondary analysis of data from the global school-based student health survey. The Lancet Child & Adolescent Health, 3(9), 616-626.
Response: added

Xu, G., Sun, N., Li, L., Qi, W., Li, C., Zhou, M., ... & Han, L. (2020). Physical behaviors of 12-15 year-old adolescents in 54 low-and middle-income countries: results from the Global School-based Student Health Survey. Journal of global health, 10(1).
Response: added

Presentation of results
- The value p must be written in lowercase and italics
Response: corrected
- In the tables, asterisks of significance could be added
*, p < 0.05; **, p < 0.01; ***, p < 0.001
Response: disagree, since the actual p-values are given
- Chi-Square test of Independence: degrees of freedom and sample size in parentheses, then the chi-square value, followed by the significance level. For example:
“Animal response to the stimuli did not differ by species, Χ2(1, N = 75) = 0.89, p = .25.”
Response: added, as below
The proportion older adolescents increased across the three different surveys X2(6, N=13,059) = 1083.26, p= <0.001) (see Table 2).
In the discussion you could compare the results with other studies in surrounding countries (see references above)
Response: Yes, this is done. However, above studies are not trend studies and therefore only included in the introduction

In the conclusion you could add proposals for social and health policies to reverse certain trends
Response: is included in the discussion, as below
The current study results may inform public health intervention programmes targeting specific health risk behaviours among adolescents in Lebanon. For example, specific school food environment policies, such as direct provision of healthful foods/beverages, can improve targeted dietary behaviours, such as fruit and vegetable intake [32]. Universal school-based interventions that target multiple-risk behaviours, may be effective in preventing engagement in substance use, including alcohol use, among young people [33]. There is “good evidence that various whole-school health interventions are effective in preventing bullying.” [34] School dental health education can improve oral hygiene practice behaviours, such as frequency and duration of brushing, of school children [35,36]. Increased implementation of multi-level (training, funding and policy) “hand-washing interventions can reduce the incidence of diarrhoea, respiratory infections, and school absenteeism.”[37] Universal resilience-focused interventions (particularly cognitive-behavioural therapy-based approaches) may be used for reductions in poor mental health (depressive and anxiety symptoms) for children and adolescents [38].

There are some mistakes:
line 156: "contolled" should be "controlled"
line 238: "respresentative" should be "representative"
Response: corrected

Round 2

Reviewer 1 Report

I thank the authors for their responsiveness to the previous comments. My primary concerns with the manuscript were addressed in this version. While I think it would be interesting to examine further trends among related risk/health behaviors (e.g., to see what behaviors might cluster together across time and show similar patterns), that seems beyond the scope of this paper. As well, it seems impossible to account for the (potential) confounding of the health promoting schools and I think the authors have acknowledged this appropriately. Aside from a need for a careful proofreading, I have no further suggestions.